# A Taxonomy of Machine Learning Clustering Algorithms, Challenges, and Future Realms

**Shahneela Pitafi \*, Toni Anwar and Zubair Sharif**

Computer and Information Science Department (CISD), Universiti Teknologi PETRONAS,
Bandar Seri Iskandar Perak 32610, Malaysia
\* Correspondence: shahneela_22000124@utp.edu.my

**Abstract:** In the field of data mining, clustering has shown to be an important technique. Numerous clustering methods have been devised and put into practice, and most of them locate high-quality or optimum clustering outcomes in the field of computer science, data science, statistics, pattern recognition, artificial intelligence, and machine learning. This research provides a modern, thorough review of both classic and cutting-edge clustering methods. The taxonomy of clustering is presented in this review from an applied angle and the compression of some hierarchical and partitional clustering algorithms with various parameters. We also discuss the open challenges in clustering such as computational complexity, refinement of clusters, speed of convergence, data dimensionality, effectiveness and scalability, data object representation, evaluation measures, data streams, and knowledge extraction; scientists and professionals alike will be able to use it as a benchmark as they strive to advance the state-of-the-art in clustering techniques.

**Keywords:** clustering algorithms; taxonomy of clustering algorithms; challenges in clustering algorithms





## 1. Introduction

Data can be categorized into numerous groups or clusters using the similarity of the data points' traits and qualities in a process known as clustering [1,2]. Numerous data clustering strategies have been developed and used in recent years to address various data clustering issues [3,4]. Normally partitional and hierarchical are the two main categories in cluster analysis approaches [5]. However, the approaches in these techniques proved incredibly effective and efficient, these methods typically rely on the availability of information on the precise dataset for every amount of clusters that need to be clustered and examined in advance [6]. Additionally, while working with an actual dataset, it is obvious to neither anticipate nor know in advance how many spontaneously existing sets there will be in the data entities.

Thus, to overcome such constraints, the idea of automatic data grouping techniques is presented. Any clustering method that automatically calculates the number of clusters without a previous understanding of the dataset's structures and qualities is referred to as using automatic clustering methods [7,8].

Many of the proposed clustering algorithms discussed in the literature and some of them are encouraged by nature, in this paper we present a review of traditional and newly offered clustering methods applied in various fields. In, the author researched the datasets occurring in statistics, computer science and machine learning. The researchers in [9,10] worked on the three V's characteristics of big data which are defined as volume, variety, and velocity which are then used in different kinds of clustering algorithms to discover. The authors in [11] presented NoPFS, a machine learning I/O middleware that eliminates the I/O bottleneck in a scalable, versatile, and user-friendly fashion. In order to take advantage of the speed and efficiency of node-local or near-node storage, the author of [12] proposed High-Velocity AI Cache (HVAC), a distributed read-cache layer. The researchers in [13] carried out brief research of the available clustering algorithms and

carried out numerous tests to identify the top clustering method for big data analysis. The authors in [14] explored data mining clustering strategies, focusing on object attribute type, scalability for huge datasets, processing high dimensional data, and identifying irregularly formed clusters. The focus of the work in [15–17] was on categorizing and summing up parallel clustering techniques. The author talked about the architecture of various parallel clustering algorithms. The writer of [15] offered a taxonomy of current clustering methods, discussing the various similarity measurements and evaluation standards for each algorithm.

Researchers in [18] conducted a comparison of the various clustering methods for both categorical and mixed datasets, and observed that for a huge dataset of any kind there is no clustering technique that can deal properly. The author of [19] observed many clustering algorithms that may be used with gene expression data in order to find and offer information on the best clustering approach that would guarantee stability and a high degree of accuracy in its analysis. The writer of [1] defined popular clustering approaches and discussed important problems and difficulties in developing clustering algorithms. The authors in [20,21] highlighted cutting-edge methods for non-numeric restrictions and big data sets of patterns.

Additionally, it can be difficult for applied researchers to discover systematic information about the subject's study growth [22]. As a result, a review of the previously conducted studies on both traditional and current clustering methods and their taxonomy is necessary to be carried out. Moreover, machine learning algorithms that are widely used in high performance and scientific domains are discussed in partitional and hierarchical clustering algorithms.

Thus, key questions for this review have been formulated as below:

1.  What are the numerous previously conducted studies on clustering approaches and techniques?
2.  How can we compare both algorithms concerning complexity and various other parameters?
3.  What are the other possible issues in clustering that still needs to be addressed?

The main contribution of this review is as follows:

A taxonomy of clustering algorithms and their brief concepts are discussed, as well as the compression of both the categories of an algorithm with various parameters. In addition, this study describes some of the most pressing questions that have arisen recently in the study of clustering concerns.

## 2. Taxonomy of Clustering Algorithms

There are two broad categories in clustering algorithms: the first is a partitional clustering algorithm and the second is a hierarchical clustering algorithm [10,15,16,18,22–26]. Agglomerative and divisive methods are further subdivisions of a hierarchical clustering algorithm. The hard or crisp clustering method, the fuzzy method, and the mixture method are three subcategories of the partitional category. There are seven main groups under the hard or crisp: search-based methods, graph-theoretic methods, density-based methods, model-based methods, sub-space methods, miscellaneous methods, and the square error as displayed in Figure 1.

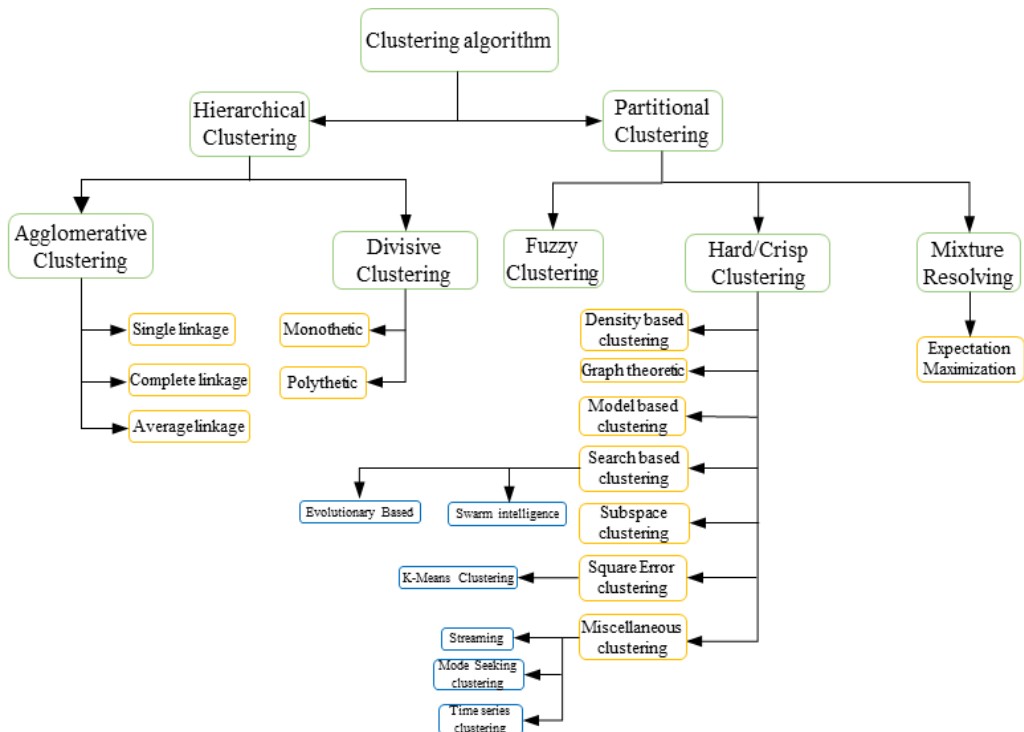

**Figure 1.** Taxonomy for various clustering techniques.

## *2.1. Hierarchical Clustering Algorithms*

Hierarchical clustering algorithms are a type of clustering algorithm where data items will be divided into sections in a hierarchical form [27]. To create a dendrogram that shows the formulated cluster's hierarchical structure, in a top-down or bottom-up way clusters are created iteratively [15]. This clustering technique enables data exploration at various granularity levels [15]. One is the divisive method where the top-down strategy is followed, and another is an agglomerative method where the bottom-up approach is followed. The process agglomerative technique follows clusters which are formed from identical items by properly combining them repeatedly into bigger clusters to establish the hierarchy's various levels. This process continues until the full object is transformed into a particular cluster or the stopping criteria is met. The opposite is true when using a polarizing strategy.

Iteratively, the cluster comprising all the objects is dispersed until either the halting requirement is satisfied, or each object creates its own cluster. The cluster element's closeness or dissimilarity is used to determine whether to merge or split the data.

The distance among the points of subgroups is calculated from the distance of individual points, hierarchy clustering allows for the merging or splitting of subsets of a point. The linkage metric, which measures proximity, is used to ascertain this. There are three kinds of linkages: one of them is single linkage, the second one is average connection and the last is complete linkage which is usually used in hierarchical clustering [15,28–31]. The algorithm for hierarchical clustering utilizes $n*n$ the linkage metrics utilized for the clustering are created in connectivity matrix form. Finding the similarities between each pair of data points allows for the building of the similarity matrix. When deciding on a linking criterion, it is common practice to measure the pairwise distance between each cluster. Using the measure of similarity, we may determine the separation between the groups of clusters. It is also utilized to answer the question of how the clusters themselves take form.

### 2.1.1. Agglomerative Clustering

In unsupervised machine learning, hierarchical, agglomerative clustering is a significant and well-established approach. Agglomerative clustering methods begin by dividing the data set into singleton nodes and gradually combining the two currently closest nodes

into a single node until only one node is left, which contains the whole data set. This process serves as a common description for several clustering systems, but they vary in how the measure of inter-cluster dissimilarity is updated after each step [32]. The objective function's optimal value serves as the criterion for selecting the pair of clusters to merge at each phase. Instead of binary data, this clustering algorithm is best suited for quantitative variables. The research of [33] devised a non-parametric hierarchy, with a conventional closest neighbor approach, an agglomerative clustering method determines a sample point's mutual neighborhood value (MNV) and mutual nearest neighbors (MNN). Agglomerative hierarchical clustering is further subdivided into the following categories.

- Single-linkage clustering: this type of clustering is also known as the minimal, connectedness, or nearest-neighbor approach. The closest distance between any two cluster members of any cluster is measured. By calculating the closest distance between a single element pair, it calculates the similarity between two clusters. The chaining effect of the single linkage clustering has the propensity to produce extended clusters [34].
- Average linkage clustering: the minimum-variance linkage is another name for average linkage clustering [15,20]. It determines the average or median distance between each cluster of data points [34].
- Complete linkage: the complete linkage, often referred to as the maximum, diameter, or farthest neighbor method, measures the longest distance between any member of one cluster and any member of the other cluster in order to calculate the distance between two clusters. Compared to single-linkage clustering, the complete-linkage algorithm clusters are smaller and more closely linked [20]. The three-proximity metrics that were described earlier take into account all the points in a pair of clusters when calculating the inter-cluster distances. They are thought of as graph techniques [10,35].

SLINK is an implementation of the single linkage hierarchical clustering technique [36]; the authors of [37,38] developed CLINK, which is an implementation of the complete linkage clustering algorithm and are examples of the average link clustering algorithm. Other geometrical techniques were created using the center point as a proximity measure based on the same concept. These comprised the minimum variance linkage metrics, centroid linkage, and median linkage metrics [39–41]. While similarity metrics capture intra-cluster connectedness, a distance-based proximity measure captures inter-cluster closeness. The adjustable amount of granularity and any similarity metric can be handled by the hierarchical clustering techniques [34].

### 2.1.2. Divisive Hierarchical Clustering

The agglomerative clustering process breaks each cluster into smaller groups starting with each item in a single cluster and continuing until the necessary number of clusters is reached and it is reversed by the process known as "divisive hierarchical clustering." The divisive approach, in contrast to the agglomerative clustering method, employs the top-down method, where the data objects are initially thought of as a fused cluster that gradually separates depending on when the cluster number is collected [42–44]. In order to divide a cluster into two subsets that each contain one or more components, the usual procedure takes into account all potential bipartitions. Even though it is common practice to examine all potential bipartitions in which each cluster is capable of being divided into two smaller clusters it is clear that the entire enumeration procedure provides a universal optimum but is quite costly in terms of computation cost.

Diverse divisive clustering methods that do not take into account all bipartitions have been researched. For instance, [45] compared the conventional K-Means or agglomerative method, and a bisecting K-Means divisive clustering method was presented. Another study [46] combined it with the divisive clustering approach to investigate a unique clustering technique dubbed "reference point-based dissimilarity measure" (DIVFRP) for the aim of dataset division.

The author in [47] proposed an improved particle optimizer (IDPSO) to identify the most convenient optimal partition hyperplane for dividing the chosen clusters into

two parts. This dividing method is a practical and effective component of the divisive hierarchical approach. The authors in [48,49] investigated the iterative division technique using the average dissimilarity between an object and a set of objects. A different strategy, however, focuses on optimization criteria that include partitioning or bi-partitioning and uses a dissimilarity matrix as input [50,51]. There are two main categories of divisive clustering: monothetic and polythetic approaches. When a set of logical qualities are both required and sufficient for inclusion in a cluster, we refer to that cluster as monothetic [52].

Monothetic divisive clusters are formed by dividing items based on a single variable in each breaking, such as whether or not they have a certain object value. The "association analysis approach" has a version called monothetic; the author of [53] developed it specifically for binary information. Several researchers have used monothetic clusters to solve problems. For instance, the authors of [54] provided an approach that gives an arrangement of things and a monothetic description of each cluster. Similarly, the author in [55] developed three monothetic techniques and principal component analysis (PCA) for inter-valued data. The initial PCA method utilized inter-valued data. The author's second approach relied on symbolic data, while their third and final algorithm was derived from the terminal values of intervals. In the end, the author tested their model using real-world data to ensure its accuracy.

Contrarily, polythetic divisive clustering is a method that uses all parameters concurrently by calculating distances or resemblance values. Rather than relying on the relative positions of variables, it relies solely on distance values, which in turn indicate the dissimilarity between all of the variables simultaneously [56].

### 2.2. Partitional Clustering Algorithm

Data is arranged into nested groups using a partitional clustering algorithm, but there is no hierarchical structure [1]. The authors in [29] claimed that applications requiring big data sets for which the creation of a dendrogram is computationally prohibitive can handle clustering problems using the partitioning method. They work by creating data clusters to recreate the natural groupings already present in the dataset. The optimization of a criterion function is used to iteratively divide the dataset of n items into a preset k number of unique subsets [57]. In [58], the author presented a new approach called k-SCC to get the best possible value for k when clustering categorical data. The squared error criteria are the most commonly used criteria functions in partitional clustering algorithms. The primary objective is to identify the segment that minimizes the square error for a given quantity of clusters. Problems in the patterns' deviations from the cluster centers are depicted by the error when the patterns are seen as a group of k-numbered spherical shape clusters. Using an information theoretic dissimilarity measure and a kernel-based technique for representation of cluster means for categorical items, the authors of [15] sought to build an unique extension of the k-means method for clustering categorical data.

The initial dataset partition serves as the basis for the partitional clustering algorithm, which allocates the data items in clusters iteratively in such a way that it can minimize the square error. The initial partition can be determined by selecting at random from the pattern matrix a set of K seed points that are widely spaced from one another. The authors of [15] emphasized the need of choosing starting points from preexisting data items that are appropriately separated from each other, appropriate seed points could be discovered. The square error tends to decrease as the number of clusters rises, and thus minimizing is only possible for a given number of clusters.

Several techniques for partition clustering employ the square error criterion function to generate K-numbered clusters that are as small and different as feasible. It is more energy efficient than alternative criteria functions [20]. Because square-error-based techniques can tend to the local optimal solution, particularly if the beginning points are not widely separated, potentially different divisions can produce various clusters as an outcome [18]. Partitional clustering is subdivided into three main categories which include fuzzy clustering, hard/crisp clustering and the last is mixture resolving.

### 2.2.1. Fuzzy Clustering

A clustering technique called fuzzy clustering was created by [59] and is based on the fuzzy set. Each pattern concurrently belongs to more than one of the fuzzy sets that form the clusters. Allocating data points to more than two clusters with varying degrees of participation in the mirrored clusters yields a non-binary relationship [22]. By allowing clusters to overlap in this manner, the fuzzy overlap is demonstrated. Fuzzy overlap counts the number of data points with meaningful membership in the overlapping clusters while reflecting the fuzziness of the cluster borders. This clustering method is useful for groupings of data points whose boundaries are fuzzy or poorly differentiated [15,49]. Information on a point's level of affiliation in a cluster might shed light on the item's inherent relationship to those clusters.

### 2.2.2. Hard/Crisp Clustering

In a hard or crisp clustering procedure, each data point refers to an exactly particular cluster. Miscellaneous clustering, density-based clustering, graph-theoretic clustering, model-based clustering and subspace clustering are some of the clustering techniques included in this category.

- Graph-theoretic clustering:

A data structure known as a "graph" is composed of nodes and the edges that connect them. When conducting data analysis, a graph can be used to list significant, pertinent features and model relationships between features of data items. Graphs are used to represent clusters in graph-theoretic clustering [15].

The representation of the data objects consists of nodes connected by edges. The edges show how close together pairs of data points are [10]. Separating cells into distinct clusters reduces the number of edges among groups while increasing the number of edges inside them [15]. Consistent edges are those whose clustering length (weight) is significantly greater than the average of the nearby edges. Points are classified into groups based on the graph structure, which results in result clusters with strong intra-connectivity/homogeneity and minimal inter-connectivity/homogeneity among the produced clusters. Although useful, representing clusters on graphs is not robust enough to handle outliers.

Using particular graph topological characteristics, this clustering method produces bunches from a collection of data items. Similar to the search for the most connected graph subgraphs, the problem of single linkage hierarchical clustering is a graph-based one. Comparable to total-linkage, hierarchical clustering is the search for the most comprehensive subgraphs in a given network [20].

- Subspace clustering:

Subspace clustering aims to discover a low-dimensional subspace that best fits each cluster of points in the data while clustering the data into numerous subspaces at the same time [60]. Instead of explaining a big dimensional dataset as a whole, it is frequently preferable to do so using the subspaces in which it exists [61]. In such a large dimensional dataset, the subspace clustering method enables the discovery of hidden knowledge. Subspace clustering makes it simple to spot clusters that exist in several overlapping subspaces. Utilizing feature selection, subspace clustering eliminates unnecessary and redundant dimensions, leaving only the pertinent dimension for the clustering algorithm to employ when locating clusters in the dataset. Using the top-down and bottom-up clustering algorithm search techniques, the subspace clustering algorithm is divided into two subcategories. The bottom-up subspace method makes use of the downward closure property of density to minimize search space using an APRORI-style methodology. Top-down subspace clustering starts by finding an estimation of the clusters throughout the full-length feature set, where each dimension is entitled to equal weight.

- Density-based clustering:

In pattern space, clusters are regarded as dense regions that are divided by less dense parts. When compared to the items in the sparse regions dividing the clusters, which are referred to as noise and outliers, the high-density regions, or modes, are connected with a cluster core [62]. The closest center clusters are then created using the data points. To determine the pattern space modes, the pattern space is divided into non-overlapping sections and a histogram is created. The valleys of the histogram's structure serve as the boundaries between the clusters, while the regions with high-frequency counts represent potential modes. The main problem with a histogram is that it requires too large of a pattern space to distinguish the portions required for determining the density function [20].

Furthermore, because they cannot be accurately described, small clusters are typically exceedingly noisy. The diverse qualities of the member patterns, however, prevent huge clusters from accurately defining the cluster properties.

Finding the exact values for the histogram's peak and the valley is another challenge [20]. Engineering has made substantial use of this clustering technique, primarily in remote sensing applications [63]. In some other instances, clusters are created depending on the number of data points present in a certain area. Data points are added to the cluster up until a predetermined threshold is reached for the neighborhood's density. In this scenario, a cluster within a specific radius must have a certain minimum number of objects that fall inside the provided criterion. Building of the cluster this makes it possible to create clusters with any shape. Naturally, noisy or outlier data points are removed. For example, density-based spatial clustering of applications with noise (DBSCAN) and optimal points for identifying cluster structure (OPTICS) are two methods for determining a clustering's organizational framework (DENCLUE) DBSCAN's cluster model is well-defined and just somewhat complicated [62]. OPTICS fixed the problem with DBSCAN's range parameter selection, producing a hierarchical outcome similar to linkage clustering [62]. Moreover, the HDBSCAN clustering algorithm is a successor of the DBSCAN algorithm; it shares all the advantages of the DBSCAN algorithm and eliminates the problem of clusters of varying densities, which is often referred as a strength of the algorithm. However, it still needs to select a minimum cluster size which is said to be its weakness [64].

According to reports, the model's complexity increased from $O(n^2)$ to $O(n\ log\ n)$ when the spatial index was used to define a data point's neighborhood [18].

- Model-based clustering:

Model-based clustering is a method for maximizing the usefulness of a selected model with the information at hand. Since clusters are formed according to the data set provided, the total nodes may be estimated rapidly, facilitating the discovery of outliers. Clustering using models utilizes a hybrid model to depict the data, with the model's components standing in for the various clusters. Cluster composition models can be generated in one of two ways, e.g., methods such as the mixing likelihood and the categorization probable. Parameter estimates for a model are often determined using the Maximum Likelihood Estimation (MLE) criteria [65], or with the Bayesian Information Criterion (BIC) [66]. The BIC may also use distance to decide which of two clusters to place a data point in [65].

- Search-based clustering:

Automatic data clustering techniques, or search-based clustering algorithms, are metaheuristic methods inspired by nature. The architecture and quantity of clusters in such a dataset are determined without prior knowledge of the characteristics or elements in the array [7]. They appear as a response to the requirement to supply the conventional clustering algorithms, a priori data [67]. Based on the quantity of clusters produced. The necessity of providing this crucial data typically results in some additional computing demands or loads on the pertinent conventional clustering techniques [7,8]. A fundamental issue in cluster analysis known as the "automated clustering problem" is finding the best estimate of the number of clusters [68]. Legitimate data clustering analysis using high-density and dimensional datasets exacerbates this difficulty. It is tough to pick sufficient cluster numbers when there is a lack of previous domain knowledge, particularly in datasets

with several dimensions and a broad range of cluster size, shape, density, and occasionally overlap. This was before the number of clusters needed for a data clustering technique is not straightforward since figuring out the appropriate number of clusters needed for such huge datasets is a very tricky subject.

For real-world data sets with high density and dimensionality, automatic clustering algorithms are developed, where such a prerequisite is not necessary, and become a superior choice. Without providing any background information about the datasets, automatic clustering algorithms yield identical results to the conventional clustering technique [7,20,22,69]. Automatic labeling of unlabeled data points in real-world datasets has also been found to be feasible using this method, which is obviously difficult and extremely difficult to accomplish traditionally.

Automatic clustering algorithms are more likely to find an optimum ideal solution than local searching algorithms, which are affected by early originating regions. A generalized optimization problem requires a linear and convex solution [7]. Additionally, nature-inspired clustering algorithms are more adaptable to addressing clustering challenges across sectors than traditional clustering approaches, which are generally a concern and lack continuity [22]. The main goal of automatic clustering algorithms is to produce clusters with lowered based on inter-distance and enhanced inter-cluster distance [7].

- Square error clustering:

Using a sum of the square error criterion functions, data points are clustered into a set number of categories with the square error clustering technique. Differentiated by the number of standard deviations, each data point from the stated group mean is included in the information shown here. When the sum of the squared errors for the data points in a cluster is zero, we may say that the points in the cluster are statistically very near to one another (very close).

K-means clustering; the K-means clustering algorithm is often used to fix clustering issues. This is an instance of unsupervised learning. Some benefits it offers are as follows: it outperforms hierarchical clustering in terms of computing efficiency for very large variables. If you choose a globular cluster with a small value for k, you will obtain denser clusters than you would with hierarchical clustering. This algorithm's main advantage is how simple it is to use and understand the clustering results. The algorithm's complexity is $O(K*n*d)$, making it very efficient from a computing standpoint [70].

Using this number, we may join the data point to the nearest cluster. When a new data point is added to a cluster, a new mean is calculated using the items already distributed among clusters, which raises the level of intra-cluster similarity. The data items are then reassigned using the new mean. Repeat this process multiple times until stability is attained. The goal of the K-Means method is to reduce the total squared error threshold [22,71,72]. The problem of the basic formulation of the number of clusters at the computation inception is one of the key concerns with K-Means clustering algorithms. There is not a reliable, all-encompassing strategy for determining how many clusters and how many partitions to start with. According to reports, the K-means algorithm is particularly sensitive to the initial centroid selection, which might lead to the production of a less-than-ideal solution [34].

- Miscellaneous clustering techniques:

Miscellaneous clustering technique includes time series mode seeking and streaming clustering techniques.

Time series clustering: time series clustering, similar to stable data clustering, needs a clustering method or technique to build clusters given a set of unordered items, and the selection of the clustering algorithm relies on both kind of information provided and the specific goal and purpose. Discrete-valued and real-valued data, uniform and non-uniform sampling, univariate and multivariate data, and data series of equal and unequal length are all various types of time series data. Before performing clustering procedures, non-uniformly sampled data must be transformed into uniform data. This may be accomplished

in several ways, from straightforward down sampling based on the roughest sampling interval to a complex modeling and estimate strategy [73].

Streaming clustering: a data stream is a vast, continually arriving series of multidimensional objects that is unlimited and quickly changing over time [74]. Since a data stream is limitless, it cannot be kept in memory or on a disc and is thus constrained to only do one pass over the data. Additionally, the order in which data arrives cannot be controlled, and objects cannot be accessed at random. Due to these limitations, traditional clustering algorithms are unable to manage the clustering issues associated with data streams [75], demanding the use of streaming clustering techniques. The number of clusters and handling outliers are two of the three main issues [74] highlighted as being unique to streaming clustering methods. Third, data streams are inherently unpredictable. In order to account for the ever-evolving aspect of streaming content, clusters produced using streaming classification techniques must be updated in real-time. A data stream is unlimited, therefore assuming a set number of clusters will be rather constrained. The cluster configuration is an ongoing process. Because both clusters and outliers change over time, it is difficult to recognize them as quickly as an object is detected a data stream. Other difficulties of streaming data that really are universal to data procedures and yet still apply to broadcast clustering include the single-stage constraints discussed above, the low computing time, which governs the fast response of the computation algorithm, and the limited computation, which also allows us to work with essential summary data.

Mode seeking clustering algorithm: the value that appears the most frequently in data collection is returned by the central tendency measure known as the mode. Both qualitative and quantitative qualities can be used to define the mode, and more than one mode may exist in a single data set. Clusters are produced using estimated density functions in mode-seeking clustering algorithms [76,77]. These modes represent the probability density functions' local maxima. Associating data samples with the closest modes in mode-seeking clustering results in the assignment of cluster labels [78]. Clusters are built automatically in mode seeking clustering technique with the number of detected modes. Mode-seeking clustering may be thought of as an agglomerative method; according to [79], every mode defines one group of data points and a density function is approximate for the dataset (every data point is used to start a new iteration of a mean shift algorithm). The density gradient from each item is tracked to determine which mode it belongs to during the clustering phase. It is said to be that a particular cluster is a place of those items that were in the same mode. According to this method, the number of clusters and the number of modes might be the same [79,80]. Both the mean shift process [79] and the K-NN mode search procedure [81] were explored in [76], both of which make use of non-parametric density estimations. The width parameter in each of the two methods affects how many modes are included in the density estimate, with clustering being considered.

### 2.2.3. Mixture Resolving Algorithms

The assumption made by the mixture-based method, also known as the mixture resolving algorithm, is that a group of observed objects originates from a combination of examples from several probability clusters. In order to produce each observed item, a probabilistic cluster is picked in accordance with the cluster's probability. After that, a sample is selected based on the selected cluster's probability density function. The data set is taken to be a combination of a certain number of distinct cluster groups that were clustered in varied proportions. The mixture likelihood-based method of clustering is model-based since it requires the pre-specification of each component's observational component density. To cluster samples from a population [82] noted that the statistical model to be employed must be specified or understood beforehand. Due to the similarity among model-based and mixture-based, simple regression concepts may be used to do prediction evaluation and hypotheses development. The mixture likelihood-based strategy, according to [82] is "probably the only clustering technique that is totally adequate from the mathematical point of view." It assumes a well-specified mathematical model, explores

it using well-known statistical techniques, and offers a test of the result's significance. A mixture-based method can readily determine the optimal number of clusters since it has a solid probabilistic foundation. Thus, according to [83], several of the advantages of the mixture model is that it successfully combines different data sets using different scientific methods. However, the processing complexity is significant, and the assumptions made about the distribution of the data are rather strong. Additionally, each cluster is seen as a single simple distribution, which limits the formation of the cluster [84].

- Expectation maximization

In data-driven methods developed, the EM technique for estimation methods has been frequently employed. EM is a statistical inference technique that guarantees the conditional probability will converge [85]. According to the parameters of the probabilistic clusters, objects are allocated to clusters in the expectation stage, and in the maximization stage, a new grouping or feature is found that improves the projected probability. Given the initial values drawn at random for the parameters of the probabilistic distribution, such as through and until the metric aligns or the shift is negligible, the average, the standard deviation, the E-step, and the M-step are repeated at regular intervals. The chance that each item belongs to each distribution is determined during clustering, and the probabilistic distribution parameter is changed to maximize the predicted likelihood of each cluster object in the M-step. The EM method requires several calculations for each iteration. As a result of this iterative calculation, the amount of data points and combination components scaled linearly, limiting the EM algorithm's applicability for large-scale applications [67,86]. The EM technique is straightforward since it does not need the setting of any factors that might influence the optimization process [67].

## 3. Comparison of Partitional and Hierarchical Clustering Algorithm

We reviewed the techniques and compared them in Table 1. The partitional methods, e.g., k-means, are generally simple, but they can only be used for certain kinds of data (convex shape). However, the clusters they create are not very reliable. However, hierarchical approaches generate very accurate clusters despite their great complexity ($O(n2)$) similarly the complexity of the partitional algorithm is $O(n(d + k))$. Furthermore, there is no global objective function for optimization in these approaches. Parameters such as k-means are used as input in many partitioning and hierarchical clustering methods and can thus impact the outcome. Results suffer if these parameters are poorly selected. Each stand-alone technique is tailored to a distinct subset of information. That is why they are so effective at handling targeted information. For some datasets, not even picking the best clustering techniques with the right settings of parameters will do. The largest datasets that contain an outlier sometimes yield poor results from even the most powerful clustering approaches. A persistent difficulty in the field of data science is the development of a clustering approach that reliably resolves all possible datasets and discovers findings with little complexity.

**Table 1.** Compression of some of partitional and hierarchical clustering algorithms.

| Categories | Algorithm/ Technique | Time Complexity | Shape of Clusters | Dataset Size | Noisy Data | Type of Dataset | High Dimensionality | Advantages | Limitations |
|---|---|---|---|---|---|---|---|---|---|
| Hierarchical based | CURE | $O(n\,2log\,n)$ | Arbitrary | Large | No | Numerical | Yes | In the first stages, it is not important to determine how many clusters will be needed. There is no need to provide any input settings. Simple to put into practice. | No room for interpretation. Extremely susceptible to extremes. Once instances are assigned to a cluster, that decision cannot be revoked. |
| | ROCK | $O(n2+nmmma+ n\,2log\,n)$ | Arbitrary | Large | Yes | Categorical | No | | |
| | BIRCH | $O(n)$ | Non-convex | Large | Yes | Numerical | No | | |
| | Ward | $O(n)$ | Non-convex | Large and small | Yes | Numerical | No | | |
| | Chameleon | $O(n2)$ | Arbitrary | Large | Yes | All types | Yes | | |
| Partitional based | K-means | $O(nki)$ | Non-convex | Large and small | No | Numerical | No | Solid, expandable, and easy to use. Simple, and it does not assume any prior experience in the field. Whenever the centroid is recalculated, other clusters form. | Challenging to forecast the number of clusters. Sensitivity to measure; this means that normalization or standards will entirely modify the results. |
| | k-medoids | $O\,(n^2kt)$ | Non-convex | Small | Yes | Categorical | Yes | | |
| | PAM | $O(k(n-k)^2)$ | Non-convex | Small | No | Numerical | No | | |
| | K-MODES | $O(n)$ | Non-convex | Large | No | Categorical | Yes | | |
| | CLARA | $O(k(m+k)^2+ k(n-k))$ | Non-convex | Large | No | Numerical | No | | |

**4. Challenges Still Exist in the Field of Clustering Algorithms**

Determining the a priori number of clusters is a significant obstacle in cluster analysis. This difficulty arises because of a scarcity of experience in the relevant field. It also happens when there are several groups inside the dataset and those groups vary in size, density, and form. Although many people have worked on this issue, it is still difficult to solve. To address this issue, future research might investigate naturally-inspired algorithms [68]. Beyond non-automatic clustering, it was hypothesized in [87] that automated clustering problems may be solved with the use of approaches inspired by nature, such as bacterium forage efficiency, fireflies' improvement, and force of gravity learning algorithms. In addition, the authors disclosed that only a small number of research ever considered hybrid nature-inspired algorithms. Traditional methods and those inspired by nature can be used to create cluster-based algorithms with improved efficiency and speed. Clustering algorithms that take cues from nature should be hybridized by combining similar algorithms in an elegant and performance-enhancing method to achieve better results. In addition, swarm intelligence-based clustering algorithms for tackling NP-hard issues in computational biology have only recently begun to be investigated [87]. Many additional unanswered questions about cluster analysis may be found in the scholarly literature, some of them are addressed in the following section.

*4.1. Computational Complexity*

While effective, certain clustering methods may be too computationally intensive to use on large-scale datasets with a high-dimensional feature map. The issue can be fixed by boosting the output of computing resources using high-capacity GPUs [88]. Moreover, it is possible that improved clustering algorithms can be designed using parallel computing to make use of the advantages it offers. Clustering methods based on parallel computing tend to be highly beneficial, but face the obstacle of complexity in implementation, as revealed by two distinct research [88,89]. As an alternative to parallel processing, MapReduce-based clustering techniques exist. Clustering techniques based on MapReduce are quicker and more scalable. To improve scalability and performance, they can implement clustering algorithms on GPU-based MapReduce frameworks.

*4.2. Refinement of Clusters*

In many cases, the clusters produced by a clustering operation need to be refined further, either using the same clustering technique or with a different clustery process. Objects that were incorrectly grouped due to ineffective similarity metrics may be relocated to the cluster where they fit best thanks to this improvement. The divisive technique is one type of clustering that uses both monothetic and polythetic approaches to the cluster refining process. The earlier method relied on the use of a single property to divide a cluster, whereas the latter method considered all available attributes. We viewed these methods as evidence that other strategies can be developed to boost cluster quality. Concerning the potential consequences of incorrectly categorizing things into clusters in potentially life-threatening contexts, this refining problem emerges as required. Hybrids of such algorithms may be considered for optimal execution of the refining work, which can increase the applicability of metaheuristic algorithms.

*4.3. Speed of Convergence*

Oftentimes, the clusters that arise from a clustering operation need to be refined further, either using the same clustering technique or with a different clustery approach. The purpose of this improvement is to move items that were incorrectly grouped owing to ineffective similarity metrics into the cluster where they belong. The divisive technique is one type of clustering that takes a two-pronged approach to the refining process of clusters, using both monothetic and polythetic means. The earlier type of cluster splitting relied on the use of a single attribute, whereas the latter type used a combination of attributes.

We viewed this as evidence that other methods can develop to enhance cluster quality. The consequences of incorrectly grouping things into clusters can have fatal consequences, making this refining issue a pressing one. In fact, hybrids of metaheuristic algorithms may be considered for optimal execution of the refining job, which would increase the application of such algorithms.

### 4.4. Data Dimensionality

Several techniques, including K-means, GMM clustering, maximum-margin clustering, and information-theoretic clustering, have trouble dealing with strong datasets. By projecting the actual information into a fairly low domain, clustering on the characteristic embedding, e.g., sparse code, can improve all of these problems [90].

### 4.5. Missing Values

During the process of collecting data from a wide variety of sources, including sensors, digital devices, machines, and people, these sources produce significant volumes of data in a very short period of time. However, collecting data is not always a simple operation, and in certain cases it might result in values being absent from the data [91]. Incomplete or missing data may obscure the true answers that lie under the surface. They are also capable of hindering the algorithms' overall performance.

### 4.6. Effectiveness and Scalability

Researchers in the field of big data clustering have a lot to learn about how to make it more effective and scalable. Deep learning has been proposed as a method for overcoming this difficulty. To further boost their efficiency, clustering algorithms might rely less on user-dependent factors. This means that in the future, researchers may consider specific needs for each area and build an algorithm that meets all of them. In addition, new clustering algorithms can be developed in the future as a result of a study into the creation of remedies for some of the fundamental difficulties of both automated and non-automatic clustering. Further study can also lead to the development of more effective algorithms to handle unexpected input without requiring a complete re-training.

### 4.7. Data Object Representation

It is also difficult for clustering algorithms to accurately describe data objects. Inappropriate data object representation is a problem. Additionally, there is a disparity in the representation of data items between domains. Data objects may be represented in a number of different ways, with some being represented as feature vectors and others as graphs with an associated concept of object similarity [9]. Distinctions in data object representation between domains of use provide a fruitful field for study. Finding an effective data representation in clustering operations is important since it improves the efficiency of clustering algorithms. By highlighting these pockets of concentrated data, the clustering technique may be made more robust and used on a larger scale. Locating sections of data that can be compressed, regions that can remain in main memory without swapping, and regions that can be ignored due to noise or lack of relevance to the results of a clustering procedure are all helpful.

### 4.8. Evaluation Measures

Different clustering algorithms may be evaluated and compared using a variety of parameters, including accuracy, algorithm stability, and dataset normalization [10]. Additionally, there is a requirement for the development of algorithmic techniques for comparing various clustering strategies with respect to several validity indices, including internal, stability, and biological indices [92]. Despite the fact that [93] highlighted that a single algorithm would not fulfill all assessment metrics, beginning with one algorithmic solution might lead to more hybridized or robust solutions.

*4.9. Data Streams*

The clustering procedure is more difficult than clustering on static data because of clustering's unique nature. Several difficulties with clustering approaches on data streams were identified by [93]. Methods of clustering should be robust enough to handle the presence of outliers and other forms of noise. To aid in the study of trends in data streams, clustering algorithms should have the ability to precisely recognize the change in context and grouping of flowing data items. Clustering algorithms' computing capabilities and memory space optimization should also increase as the number of data streams generated from various sources, such as social networks, grows. To further adapt current context-based adaptive clustering algorithms and develop models for clustering dynamic data streams, more study is needed.

*4.10. Knowledge Extraction*

Clustering also faces difficulties in extracting useful information from large datasets. The rise in data creation and storage is to blame [22]. Terabytes and petabytes of data provide a significant hurdle for the data analyst because of this issue. Knowledge extraction from large datasets has limitations that can be solved with more research. Current methods, such as distributed clustering and parallel evolutionary algorithms, need improvement. In addition, future research can build novel clustering algorithms that can pick and decide among single-objective and multi-objective optimizations.

## 5. Conclusions and Future Directions

Clustering has found widespread use in data mining and analysis across several disciplines, computer science, data science, statistics, pattern recognition, artificial intelligence, and machine learning, and so on. The most basic issue in cluster analysis is determining the number of clusters beforehand. Many clustering issues may be solved more effectively if the right number of clusters is specified in advance. This is why automated clustering methods are becoming the norm rather than the exception. To perform clustering without needing any prior knowledge of data sets, automated clustering techniques were developed. They can also determine how many groups should be included in a noisy dataset. This research provides a thorough and current overview of both classic and cutting-edge clustering techniques. Both researchers and practitioners may learn from this study.

Furthermore, future research might examine the qualities of various clustering techniques and how both maps efficiently tackle problems in other application fields, as well as the uniqueness of the difficulties faced in those sectors. Furthermore, future research might look at the use of newer or even hybrid clustering algorithms in a chosen subject based on the numerous tendencies highlighted in this study. Researchers and professionals can use the information in this study as a starting point for developing new, more effective, and efficient clustering algorithms.

**Author Contributions:** Conceptualization, S.P.; methodology, S.P.; software, S.P.; validation, Z.S.; formal analysis, Z.S.; investigation, S.P.; resources, S.P.; data curation, S.P.; writing—original draft preparation, S.P.; writing—review and editing, Z.S.; visualization, Z.S.; supervision, T.A.; project administration, T.A.; funding acquisition, T.A. All authors have read and agreed to the published version of the manuscript.

**Funding:** This research work is supported and funded by Universiti Teknologi PETRONAS (UTP) under the YUTP grant scheme with the cost center of 015LC0-350.

**Institutional Review Board Statement:** Not applicable.

**Informed Consent Statement:** Not applicable.

**Data Availability Statement:** Not applicable.

**Acknowledgments:** The authors would like to thank Universiti Teknologi PETRONAS (UTP) for all the support provided for this research work.

**Conflicts of Interest:** The authors declare no conflict of interest.

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
