# Peer review of "A Taxonomy of Machine Learning Clustering Algorithms, Challenges, and Future Realms"

_applsci, doi:10.3390/app13063529_

Round 1

Reviewer 1 Report

The authors conducted a through review of existing articles and describe in detail existing studies in detail. 

- Strength: The work by authors is comprehensive and timely. 

Weakness: 

- I do not see any key contributions by authors in terms of concluding an analysis of existing studies based on defining certain metrices etc. Thus, the draft gives an impression that authors did summarized existing studies and provided a simple review.

- I was expecting some experimental analysis of top rated techniques but was also disappointed by the fact that authors didn't included any of such material. I highly suggest authors to include such elements, as it would make the manuscript highly impactful.

- Several recent references are missing from top CS conferences, e.g., 1. Clairvoyant NOPFS (SC'21), 2. Removing i/o bottleneck for large-scale deep learning applications (Cluster'22) and several others. 

- Also please add ML algorithms being used highly in HPC and scientific domains.

Reviewer 2 Report

This paper introduces a taxonomy of recent clustering algorithms. It visits some classical and SOTA hierarchical and partitional clustering algorithms with various parameters. It also discusses some open challenges in clustering such as computational complexity, refinement of clusters, speed of convergence, data dimensionality, effectiveness and scalability, data object representation, evaluation measures, data streams, and knowledge extraction.

In general, the survey is well-written and organized. The topic is still suitable for the scope of AS journal since clustering can be applied in many fields and as preliminary steps in other works. This paper is easy-to-read, even with non-expert in this field. In supporting this paper, the authors should consider the following points to improve the performance of the paper further.

- In the Introduction, highlight the originality of the paper. As you can see, we may find many survey papers/posts on clustering. what are the most interesting points the potential readers can find in this paper?

- In the Introduction, the authors should give some info on several other clustering methods. I have used k-means, HAC, DBSCAN, and HDBSCAN- an improved version of DBSCAN that allows varying density clusters instead of using a global epsilon distance as in DBSCAN. I observed that k-means, DBSCAN, and HDBSCAN could perform very well in the clustering task for large-scale datasets. In some cases, DBSCAN and HDBSCAN were even better than k-means since they can remove noises. I also used the HDBSCAN python version. It is an efficient algorithm in terms of runtime and can work well with high-dimensional data. Thus, the authors should also summarize/compare the strength and weaknesses of HDBSCAN [https://hdbscan.readthedocs.io/en/latest/comparing_clustering_algorithms.html].

- In section 2, the authors should update some recent works in clustering in the taxonomy.

- In section 2.2, I prefer if the authors discuss the partitional clustering algorithm for different types of data, such as categorical data and mixed data since they are very common in real-life applications. The author can mention the paper ''A method for k-means-like clustering of categorical data" and "Estimating the Optimal Number of Clusters in Categorical Data Clustering by Silhouette Coefficient" in the discussion.

- In section 4, I prefer if the authors discuss the problem of missing values in clustering. Missing values may hide the correct answers underlying the data. They can also reduce the performance of the algorithms.

- In section 5, since this is a survey, it is better if authors give some opportunities for researchers/scholars working on clustering.

- Proofread the paper to fix all typos, for example, "[35,36]      developer", or  "[61]Cluster", "[77]and"

Round 2

Reviewer 1 Report

Authors have addressed all of my concerns.

Reviewer 2 Report

I have checked this revision. The authors have considered all my comments/concerns and improved the paper accordingly. The quality of the paper is fine. Thus, I recommend acceptance.